# Improvement of Vascular Insulin Sensitivity by Ranolazine

**DOI:** 10.3390/ijms241713532

**Published:** 2023-08-31

**Authors:** Sol Guerra-Ojeda, Adrian Jorda, Constanza Aldasoro, Jose M. Vila, Soraya L. Valles, Oscar J Arias-Mutis, Martin Aldasoro

**Affiliations:** 1Department of Physiology, University of Valencia, 46010 València, Spain; solanye.guerra@uv.es (S.G.-O.); adrian.jorda@uv.es (A.J.); constanza.aldasoro@gmail.com (C.A.); vila@uv.es (J.M.V.); lilian.valles@uv.es (S.L.V.); oscarariasphd@gmail.com (O.J.A.-M.); 2Department of Nursing and Podiatry, University of Valencia, 46010 València, Spain

**Keywords:** insulin, ranolazine, nitric oxide, vascular, adrenergic system

## Abstract

Ranolazine (RN) is a drug used in the treatment of chronic coronary ischemia. Different clinical trials have shown that RN behaves as an anti-diabetic drug by lowering blood glucose and glycosylated hemoglobin (HbA1c) levels. However, RN has not been shown to improve insulin (IN) sensitivity. Our study investigates the possible facilitating effects of RN on the actions of IN in the rabbit aorta. IN induced vasodilation of the abdominal aorta in a concentration-dependent manner, and this dilatory effect was due to the phosphorylation of endothelial nitric oxide synthase (eNOS) and the formation of nitric oxide (NO). On the other hand, IN facilitated the vasodilator effects of acetylcholine but not the vasodilation induced by sodium nitroprusside. RN facilitated all the vasodilatory effects of IN. In addition, IN decreased the vasoconstrictor effects of adrenergic nerve stimulation and exogenous noradrenaline. Both effects were in turn facilitated by RN. The joint effect of RN with IN induced a significant increase in the ratio of p-eNOS/eNOS and pAKT/AKT. In conclusion, RN facilitated the vasodilator effects of IN, both direct and induced, on the adrenergic system. Therefore, RN increases vascular sensitivity to IN, thus decreasing tissue resistance to the hormone, a key mechanism in the development of type II diabetes.

## 1. Introduction

IN is a peptide hormone secreted by the beta cells of the islets of Langerhans that is the most important hormone in anabolic functions regulating glucose homeostasis. In addition to its metabolic effects, its cardiovascular actions are of special relevance, among them the synthesis of nitric oxide (NO) [1], endothelin 1 (ET-1) [2], or its effects on the adrenergic system [3].

IN has vasodilatory effects that are induced by the release of endothelial NO [1]. On the other hand, it induces the release of endothelial hyperpolarizing factor (EDHF), which generates hyperpolarization of the cell membrane, stimulates the K^+^-Na^+^ ATPase pump, and activates Ca^2+^-dependent K^+^ channels, thus reducing the entry of Ca^2+^ through voltage-channel dependents [4]. The hormone also exerts its action by activating the Ca^2+^ pumps of the membrane and the sarcoplasmic reticulum, thus decreasing cytoplasmic Ca^2+^ concentrations [5]. In both ways, either by stimulating the exit of Ca^2+^ from the cytosol towards the sarcoplasmic reticulum or by preventing the entry of extracellular Ca^2+^ into the cytosol, relaxation of the vascular smooth muscle is produced, which leads to vasodilation. Furthermore, through MAPK, IN induces the formation and synthesis of endothelial ET-1, which generates vasoconstriction, opposing the vasodilator effect of NO [2].

IN resistance, or the dysfunction in response to hormones, could generate the pathological alterations characteristic of metabolic syndrome and type 2 diabetes mellitus as part of this syndrome. This hypothesis suggests that certain key effects of IN such as its vasodilator action, would be progressively reduced, while other effects regulated by the other IN signaling pathway, such as those that promote vasoconstriction, vascular hypertrophy, or sympathetic nervous system potentiation, would be potentiated [6]. The cardiovascular system itself is the most sensitive to the effects of IN and the blood vessels are the tissue in which the effects of IN resistance begin [7]. The imbalance between its stimulatory and inhibitory cardiovascular actions would lead to the appearance of hypertension and other disorders that accompany metabolic syndrome, such as atherosclerosis, endothelial dysfunction, or tissue ischemia [8].

RN selectively inhibits the late inward Na^+^ current (INaL), reduces intracellular Na^+^ accumulation and subsequent Ca^2+^ accumulation induced by the Na^+^-Ca^2+^ exchanger, as well as mechanical, electrical, and metabolic abnormalities in the ischemic myocardium [9,10]. In various clinical trials conducted in patients with chronic stable angina, RN exerts antianginal and antiischemic actions, and in patients with acute coronary syndrome, it exerts antiarrhythmic actions [11,12,13]. RN is also used in the treatment of arrhythmias [14], being a drug of special interest in atrial fibrillation [15] or different types of heart failure [16], both systolic and diastolic. Its beneficial effect has also been postulated in diastolic dysfunction induced by cardiac oxidative stress [17] as well as in different types of angina, such as microvascular [18], and its therapeutic role has recently been demonstrated not only in microvascular angina but also in severe refractory angina [19], as well as in the treatment of arrhythmias [20] and hypertrophic cardiomyopathy [21], being more effective than beta-blockers [22]. On the other hand, different benefits of the drug have been revealed in type II diabetic patients, in whom, in addition to its cardiac effects, it decreases blood glucose and glycosylated hemoglobin (HbA1c) levels [23,24], improving cognitive aspects in patients with type 2 diabetes mellitus [25]. Different multicenter studies have demonstrated the antidiabetic effects of the drug. In type 2 diabetic patients, RN decreased glycosylated hemoglobin (HbA1c) by 7% [26,27] and reduced the fasting glucose by 25.7 mg/Dl [28], although they are not conclusive about its possible intervention in the sense of facilitating the peripheral effects of IN by reducing tissue resistance to the hormone [28]. An alternative site of action for RN, through opening sodium channels, causes a decrease in the release of glucagon by pancreatic alpha cells, which improves both postprandial and preprandial glucose levels [28].

Therefore, the objective of this study is to analyze the effects of IN in the rabbit aorta (the common arterial vessel that initiates the entire systemic circulation) and the facilitating mechanisms of RN on vascular sensitivity to IN. In this sense, it is intended to assess the effects of RN on the AKT-eNOS IN signaling pathway and on the vascular adrenergic system, since IN resistance begins in the vascular wall.

## 2. Results

### 2.1. Response of the Rabbit Aorta to IN

IN-induced concentration-dependent relaxation of rabbit aortic segments. The maximum effect was 48% relaxation, and it was completely blocked when the arterial segments were incubated with S961, an IN-receptor antagonist, which indicates that vasodilation due to IN is mediated by its membrane receptor (Figure 1A) (Table 1). Incubation with L-N^G^-Nitro arginine methyl ester (LNAME), which blocks NOS and therefore NO synthesis, induces a 45% block of the relaxing effect mediated by IN (Figure 1B) (Table 1).

On the other hand, incubation with tetraethyl ammonium (TEA), an endothelium-derived hyperpolarizing factor (EDHF) blocker, did not change the vasodilator response induced by IN (Figure 2) (Table 1). Neither were changes observed in the responses to IN after incubation with indomethacin (INDO) (10^−6^ M), which blocks the synthesis of prostanoids, both vasodilators and vasoconstrictors (Figure 2) (Table 1). Therefore, approximately half of the relaxing response is produced by NO release, while the rest of the relaxation has not been able to determine its original mechanisms.

IN induces the release of ET-1 via the MAPK pathway. In our results, incubation with BQ123 (10^−6^ M), an endothelin A (ETA) receptor blocker, or with BQ788, an endothelin B (ETB) receptor blocker, did not induce significant changes in the vasodilator response generated by IN, so ET- 1 does not seem to play a role in these responses after binding to ETA receptors, which induce vasoconstriction, or to ETB receptors, which mediate vasodilation (Table 1).

IN (10^−9^ M) caused a shift to the left of the concentration-response curve to acetylcholine, shifting the maximum vasodilator effect from 75% to 90% (Figure 3) (Table 2).

IN did not cause significant changes in the relaxation induced by sodium nitroprusside (NPS) (Figure 4) (Table 3).

### 2.2. Effects of RN on the Vasodilator Responses due to IN

Incubation with RN within the therapeutic range (10^−5^ M) caused a significant increase in the relaxing response induced by IN that reached a 74% maximum effect (Figure 1C) (Table 1).

On the other hand, RN 10^−5^ M increased the relaxing response induced by acetylcholine, both pD2 and the maximum effect, as well as the facilitating effect of IN on acetylcholine with an increase in pD2 without changes in the maximum effect (Figure 3) (Table 2).

As already indicated, IN did not cause significant changes in the relaxation induced by sodium nitroprusside, neither in the pD2 values nor in the maximum effect. In contrast, the presence of 10^−5^ M RN induced a shift to the left of the concentration-response curve to NPS with a significant increase in pD2 both compared to the control curve and the curve after incubation with IN, with no differences in the maximum effect in either case (Figure 4) (Table 3).

### 2.3. Effects of IN and RN on the Response of the Aorta to Sympathetic Nerve Stimulation

IN (10^−9^ M) caused a statistically significant reduction in the vasoconstrictor response induced by sympathetic nerve stimulation at 2, 4, and 8 Hz. Subsequent incubation with RN 10^−5^ M caused further IN blockade of the vasoconstriction induced by the sympathetic nervous stimulation in all the stimulus frequencies studied (Figure 5). On the other hand, prazosin (PRZ) (10^−4^ M) induced a competitive antagonism of alpha-1 adrenergic receptors, demonstrating the viability of these receptors in the vascular response to adrenergic stimulation (Figure 5).

### 2.4. Effects of IN and RN on the Response of the Aorta to Noradrenaline

Noradrenaline induces concentration-dependent vasoconstriction in segments of the rabbit abdominal aorta. The presence of IN (10^−9^ M) causes a shift of the concentration-response curve to noradrenaline to the right. Subsequent incubation with RN (10^−5^ M) caused a significantly greater curve shift than IN. Incubation with prazosin (10^−4^ M), as an antagonist of alpha-1 adrenergic receptors, shifted to the right in a parallel form of the curve concentration-response to adrenaline, acting as a competitive antagonist of these receptors (Figure 6) (Table 4).

### 2.5. Effects of IN and RN on eNOS, p-eNOS, AKT and p-AKT Protein Expression

IN (10^−9^ M) did not produce changes in the expression of the p-eNOS/eNOS ratio compared with the control, and no changes were detected when the aortic segments were incubated with IN (10^−9^ M) and RN (10^−6^ M). On the other hand, when the RN concentration used was 10^−5^ M (together with IN), there was a statistically significant increase in the p-eNOS/eNOS ratio, both with respect to the control and to that obtained with IN alone (Figure 7).

Furthermore, IN (10^−9^ M) did not induce changes in the expression of the p-AKT/AKT ratio compared with the control, and no changes were observed when incubated together IN (10^−9^ M) and RN (10^−6^ M). However, with RN 10^−5^ M (together with IN) a significant increase in the p-AKT/AKT ratio was detected compared both with control and with IN alone (Figure 7).

## 3. Discussion

In our study, the functional capacity of the rabbit aorta, as observed by the vasoconstrictor response induced by KCl, was similar to that obtained in different vascular territories [29,30]. Our results demonstrate that IN induces concentration-dependent relaxation of the rabbit abdominal aorta by binding to specific membrane receptors (Figure 1) (Table 1). This relaxation is mainly due to the synthesis of NO (Figure 1), coinciding with other previous results [1,31], although in our study NO was responsible for 45% of the relaxation induced by IN. The rest of the vasodilator effect of IN is due to other factors that we have not been able to determine since it is not the EDHF factor or prostanoids because the relaxation did not vary with the use of TEA, EDHF synthesis blocker, or indomethacine (prostanoid synthesis inhibitor) (Figure 2) (Table 1). The relaxing effect reached a maximum of 40% of the contraction due to noradrenaline and was totally inhibited by the specific IN receptor antagonist S961 (Figure 1). The response was entirely dependent on the endothelium, since it was not produced in arterial segments devoid of this cell layer. Other mechanisms have been proposed that would explain the vasodilator effect of IN, such as the modulation by the hormone cytosolic Ca^2+^ [32], the stimulation of Na^+^ ATPase, or the activation of Ca^2+^-dependent K^+^ channels, thus reducing the entry of Ca^2+^ [4]. On the other hand, IN potentiated the vasodilatory effects of acetylcholine (Figure 3) (Table 2) but not those produced by NPS (Figure 4) (Table 3). Thus, it seems clear that IN facilitates acetylcholine-induced NO synthesis but not vascular smooth muscle sensitivity to NO since NPS is an NO donor. An essential feature of IN resistance states consists of the impairment of the phosphoinositide 3-kinase (PI3K)-dependent intracellular signaling pathway and, consequently, of the formation of endothelial NO [33]. This point is key to the development of vascular resistance to IN. Other signaling pathways, such as Ras/MAPK and ET-1 formation, are not affected [34]. However, the increase in the production and biological activity of ET-1 has also been proposed as a trigger for endothelial dysfunction secondary to vascular resistance to IN [35]. Any change in the balance between the vasodilator and vasoconstrictor actions of IN can cause resistance to the hormone and endothelial dysfunction [36,37]. ET-1 decreases IN-stimulated capillary recruitment and glucose uptake in different muscle cells [32,38]. However, since NO inhibits the production and actions of ET-1 [39] under physiological conditions, the actions of ET-1 produced by IN are balanced by the production of NO [32].

RN potentiated the vasodilator effects of IN (Figure 1C). This effect of RN probably depends on an increase in eNOS phosphorylation (Figure 7), as observed in protein expression studies, and on the consequent increase in NO synthesis. We have also observed that RN potentiated the facilitating mechanism of IN on the vasodilation induced by acetylcholine (Figure 3), as well as the effect of IN on NPS and, therefore, the sensitivity of smooth muscle to NO (Figure 4) (Table 3). RN seems to facilitate the synthesis of endothelial factors such as NO [40], although other authors do not associate its vascular effects with the release of endothelial factors [41,42]. Studies in diabetic patients seem to indicate that RN promotes endothelial function by stimulating eNOS [43], although other ways by which RN facilitates endothelial function have been suggested, such as its antioxidant and anti-inflammatory effects [39]. A possible mechanism by which RN facilitates the vasodilator effect of IN, demonstrated in coronary ischemia, is the inhibition it exerts on the beta-oxidation of fatty acids that would increase coronary blood flow [9,44]. RN facilitates ATP production and oxygen consumption by stimulating glucose oxidation and decreasing fatty acid oxidation [44]. Different effects of RN have been demonstrated, such as a decrease in blood glucose and glycosylated hemoglobin levels [45], the promotion of IN release, and a decrease in glucagon synthesis [46], thus improving pre- and postprandial glycemia [26,28]. RN has been shown to be useful in preventing cognitive decline in patients with type II diabetes [25]. Likewise, its clinical use may be interesting in patients with type II diabetes and coronary ischemia [47,48], and it has even been proposed as the first treatment in patients with type II diabetes [27]. RN does not modify the AKT pathway or the kinases involved in glucose uptake [49]. In our study, RN facilitated the effects of IN on AKT (Figure 7), coinciding with previous studies in our laboratory performed on primary cultures of astrocytes [50]. RN decreases cellular resistance to IN in diabetic patients with coronary disease, improving the HOMA-IR index with more promising results than those obtained with beta-blockers or Ca^2+^ channel blockers [51]. In any case, there is no direct evidence that RN increases cellular sensitivity to IN. However, our data indicate that RN increases vascular IN sensitivity and does so by facilitating eNOS function with the consequent increase in NO synthesis, but also by increasing smooth muscle sensitivity to NO.

In our study, we analyzed the effects of sympathetic nervous stimulation on the rabbit aorta by using electrical field stimuli (EFS) capable of causing the release of specific neurotransmitters from the sympathetic nervous system. The response was abolished in the presence of tetrodotoxin (TTX), Guanethidine, or Prazosin. IN inhibited the contractile response to adrenergic nerve stimulation at all stimulation frequencies used (Figure 5). This inhibitory effect was, in turn, potentiated by RN at all the frequencies studied (Figure 5). Some studies show that RN inhibits the contractile effects of adrenergic agents [42], as well as having a partial inhibitory effect on adrenergic nerve stimulation [52]; however, the possible interactions between RN and IN in the vascular territory have not been described. In our study, noradrenaline, an adrenergic nervous system neurotransmitter, produced concentration-dependent contractions in the aortic segments (Figure 6) (Table 4). IN induced a statistically significant right shift of the concentration-response curve to noradrenaline (Figure 6) (Table 4). Incubation with RN potentiated the effect of IN, generating a greater shift of the concentration-response curve to the right (Figure 6) (Table 4). The affinity of noradrenaline in the rabbit aorta was similar to that obtained in different vessels, such as the human middle cerebral artery [53], and lower than that in cutaneous vessels [54]. An antagonistic effect of RN on adrenergic function has been suggested after inhibiting postsynaptic alpha-1 receptors [42]. An interesting aspect is the participation of the adrenergic nervous system in the prevention of oxidative stress by increasing intracellular levels of GSH, an effect mediated by beta-3 receptors [55,56], also implicated in the prevention of vascular damage in the development of arterial hypertension [57]. RN decreases the oxidative stress involved in multiple myocardial mechanical dysfunctions and different types of arrhythmias, either by modulating the formation of reactive oxygen species (ROS) [17,58] or by facilitating the function of different superoxide dismutase enzymes (MnSOD and Cu/ZnSOD) [59,60]. On the other hand, the hypofunction of the PPAR-gamma transcription factor, which inhibits the inflammatory process [61], has been implicated in the development of vascular resistance to IN. In a previous study in our laboratory [59], RN increased the expression of PPAR-gamma in cell cultures. Therefore, it is possible that this mechanism could be involved in the facilitation of the vascular effects of IN by the RN.

In conclusion, this study suggests that RN, at therapeutic concentrations, facilitates the vasodilatory effects of IN in the rabbit aorta, mediated through the AKT and eNOS pathways, as well as the effects of IN on acetylcholine-induced vasodilation, and increases smooth muscle sensitivity to IN. In addition, RN facilitates the inhibition of IN on vasoconstriction of adrenergic origin, both due to sympathetic nervous stimulation and that induced by the direct effect of noradrenaline. The various drugs used in the control of type II diabetes normally focus on increasing the release of IN from the pancreas, thus lowering blood glucose and glycosylated hemoglobin. However, there are no drugs for antidiabetic use that are capable of reducing vascular resistance to IN, the origin of type II diabetes. Therefore, RN could be a useful drug in the treatment of this type of diabetes.

## 4. Materials and Methods

### 4.1. Animal Model

The present study was carried out in accordance with the ethical standards in animal experimentation established by EU Directive 2010/63 and Spanish Royal Decree (RD) 1201/2005. The experiments were developed using shared tissue samples from procedure 2017/VSC/PEA/00049 type 2, authorized by the Bioethics Committee of the University of Valencia, Spain. Twenty-nine male New Zealand white rabbits weighing 2.8–3.6 kg was used in this study and housed in a 12:12 h light/dark cycle at a constant room temperature of 22 °C and 60% humidity. Animals were euthanized following heparinization and anesthesia (sodium thiopental 60 mg/kg i.v.).

### 4.2. Preparation of Vascular Rings

In vitro organ bath experiments were carried out as previously described [62]. After pharmacological euthanasia, the abdominal aorta was isolated and cut into 4-mm rings for isometric recording of tension. Two stainless steel L-shaped pins were introduced through the lumen of the aortic rings. One pin was fixed to the wall of the organ bath, and the other one was connected to a force-displacement transducer (FT03; Grass Instruments, West Warwick, RI, USA). Variations in isometric force were registered on a Macintosh computer (Apple Corp., Cupertino, CA, USA) using the Chart, version 7, and a MacLab/8e data acquisition system (AD Instruments). Individual rings were suspended in a 5 mL bath with a modified Krebs-Henseleit solution containing (mM) NaCl, 115; KCl, 4.6; CaCl_2_, 2.5; MgCl_2_•6H_2_O, 1.2; NaHCO_3_, 25; glucose, 11.1; and disodium EDTA, 0.01, with 95% O_2_ and 5% CO_2_ to obtain a pH of 7.3–7.4 at 37 °C. The optimal resting tension for aortic rings was 3.5 g, and vascular preparations were allowed to equilibrate for 2.5 h. The contractile capacity of aortic smooth muscle was evaluated by its response to KCl (60 mM). The endothelium was considered functional if immediate relaxation to a single concentration of acetylcholine (10^−6^ M) in rings precontracted with noradrenaline was ≥70%. Aortic rings with dysfunctional endothelium in the control conditions were excluded.

### 4.3. Experimental Procedure

To assess the possible interactions of IN with different vasoactive agents, concentration-response curves were performed in which the aortic segments were contracted with noradrenaline (10^−8^ to 3 × 10^−6^ M). After reaching a stable contraction of the vascular segments, IN was added to the organ bath in increasing and cumulative concentrations (10^−12^–10^−8^ M), and then the vasodilator responses to the hormone were recorded. The intervention of IN membrane receptors in these responses was analyzed by incubating (for fifteen minutes) the vascular segments with S961 (10^−6^ M), a specific antagonist of these IN receptors, and then constructing new concentration-response curves for the hormone.

The intervention of nitric oxide in the vascular responses to IN was observed by incubating the aortic segments with the eNOS antagonist, NG-nitro-L-arginine methyl ester (L-NAME 10^−4^ M), together with IN (10^−12^–10^−8^ M), for fifteen minutes. The intervention of prostanoids, either contractile or relaxant, was tested by incubating the aortic segments with the COX_1_ and COX_2_ inhibitor, indomethacin (INDO) (10^−6^ M) (for fifteen minutes). To assess the role of hyperpolarizing factor of endothelial origin (EDHF), the aortic segments were incubated for fifteen minutes with TEA (10^−4^ M), a K^+^ channel blocker, the target point of the EDHF mechanism of action. ET-1 is part of the vasoactive agents synthesized by the effect of IN. In aortic segments, different from those used for the control curves, we incubated with the endothelin receptor type A (ETA) receptor blocker, BQ123 (10^−6^ M), fifteen minutes before starting the concentration-response curves to IN as previously indicated.

In order to assess the effects of RN on the vascular action mechanism of IN, segments of rabbit aorta, not subjected to previous treatments, have been incubated independently with RN (10^−5^ M) (for fifteen minutes) as shown in Figure 1, performing the concentration-response curves to IN (Figure 1). After achieving stabilization of the aortic segments at optimal and constant basal tension, concentration-response curves were obtained for two vasodilator agonists, specifically acetylcholine and sodium nitroprusside. The aortic segments were contracted with noradrenaline in submaximal doses (10^−7^–3 × 10^−6^ M). The administration of the vasodilator agonists was carried out cumulatively, so that, when applying a dose, the concentration of the agonist in the aqueous medium of the organ bath results from the sum of the latter with the concentrations reached previously. Subsequent changes in concentration were obtained as the previous ones reached their maximum effect.

After reaching a stable contraction, concentration-response curves for acetylcholine (10^−9^–10^−6^ M) were constructed. Subsequently, the aortic segments were incubated with IN (10^−9^ M). After fifteen minutes of incubation, the concentration-response curves for acetylcholine (10^−9^–10^−6^ M) were repeated. After characterizing the effects of IN on acetylcholine, the possible facilitating effect of RN on IN actions was analyzed. The aortic segments were incubated together with IN (10^−9^ M) and RN (10^−5^ M) to perform the concentration-response curves for acetylcholine again after fifteen minutes.

Due to the NO donor condition of sodium nitroprusside, the analysis of its vasodilator effect is aimed at determining the sensitivity of the vascular wall to NO. Aortic segments were incubated with IN (10^−9^ M) (for fifteen minutes) and concentration-response curves to sodium nitroprusside were performed. After determining the nature of the interaction of IN with sodium nitroprusside, we proceeded to analyze the possible facilitation of RN on IN. With this objective, aortic segments, different from those of the previous experiment, were incubated at the same time with IN (10^−9^ M) and RN (10^−5^ M) to perform the concentration-response curve to sodium nitroprusside after fifteen minutes.

### 4.4. Periarterial Adrenergic Nerve Stimulation

To obtain adrenergic nerve stimuli, electrical field stimulation (EFS) was applied through two platinum electrodes placed on both sides of the aortic vascular segment with a 5-mm separation between the two electrodes. The electrodes were connected to a multichannel stimulator (Grass S88). The correspondence between frequency and vasomotor response was studied in a certain range of frequencies, specifically 2, 4, and 8 Hz, with the application of 25 V stimuli (considered supramaximal voltage) of 0.25 ms duration for each pulse for 30 s of total duration of stimulation.

The analysis of the neurogenic nature of the contractile response to EFS was carried out by incubating the aortic segments for 15 min with TTX (10^−6^ M), an inhibitor of voltage-dependent Na^+^ channels and, therefore, a blocker of the nerve conduction of the neural fibers present in the vascular wall; guanethidine (10^−6^ M), an antagonist of the release of noradrenaline, a specific neurotransmitter of the adrenergic nervous system; therefore, it also blocks adrenergic neurotransmission; or with a receptor antagonist alpha-1 adrenergic postsynaptic drugs, prazosin (10^−4^ M).

In each experimental session, different stimulation rounds (2, 4, and 8 Hz) were provoked, with 5 min intervals between each stimulus of increasing frequency. These series of stimuli were applied again 10 min after adding TTX, guanethidine, or prazosin to the organ bath. As a control group, in another series of aortic segments, another series of electrical field stimuli was performed without the presence of adrenergic nervous system blockers.

### 4.5. Interaction between IN and RN with the Endogenous Adrenergic System

The possible interaction of IN with the sympathetic-adrenergic nervous system was assessed by incubating the aortic segments with IN (10^−9^ M) for fifteen minutes prior to the application of the sequence of adrenergic nervous stimuli at increasing frequencies (2, 4, and 8 Hz). The interaction between RN and IN was revealed by joint incubation of the aortic segments with IN (10^−9^ M) and RN (10^−5^ M), followed by a sequential series of adrenergic nerve stimuli at the frequencies referred to above.

### 4.6. Concentration-Response Curves to Noradrenaline

After stabilizing the aortic segments at optimal basal tension values, we proceeded to design the concentration-response curves for noradrenaline administered in an accumulative form (10^−9^–10^−5^ M), so that the concentration of the agonist in the organ bath when applying successive doses is the result of the sum of the current dose with those previously administered. The following dose was administered when the maximum contractile effect was reached with the previous one. Subsequently, the different blockers, antagonists, or agonists were administered to the organ bath, normally for 15 min, before obtaining the corresponding specific concentration-response curves for each treatment. To obtain the control group, concentration-response curves were designed in other aortic segments without the presence of blockers, antagonists, or agonists.

### 4.7. Interaction between IN, RN, and Exogenous Noradrenaline

The analysis of the effects of IN on the vascular responses to noradrenaline of rabbit aortic segments was performed by incubating the segments with IN at a concentration in the organ bath of 10^−9^ M for fifteen minutes, then proceeding to obtain the different concentration-response curves for noradrenaline. As in previous experiments, another series of aortic vascular segments was used as a control group in which concentration-response curves to noradrenaline (10^−9^–10^−5^ M) were carried out without the presence of IN. After analyzing the interactions between IN and exogenous noradrenaline, we assessed the effects of RN on the relationship between IN and noradrenaline. For this, the rabbit aortic segments were incubated together with IN (10^−9^ M) and RN (10^−5^ M) to obtain, from these data, the concentration-response curves to noradrenaline (10^−9^–10^−5^ M).

### 4.8. Western Blot Analysis

Aortic rings for protein expression were incubated in the organ bath for 20 min in control conditions and with IN (10^−9^ M), RN (10^−6^ and 10^−5^ M), and stored in liquid nitrogen at −80 °C until analysis.

Aortic tissues were homogenized in lysis buffer (0.125 M Tris–HCl, pH 6.8, 2% SDS, 19% glycerol, and 1% *v*/*v* protease inhibitors) and centrifuged at 12,000× *g* for 15 min at 4 °C. Protein concentration was determined using the BCA method (Thermo Scientific Pierce BCA Protein Assay Kit, Abcam, Boston, USA). Furthermore, 0.5% (*v*/*v*) 2-mercaptoethanol and 1% bromophenol blue were added, and the samples were heated for 5 min at 90 °C. Proteins (20 µg) were separated on SDS–PAGE gels and transferred to polyvinylidene difluoride membranes in a humid environment using a transfer buffer (25 mM Tris, 190 mM glycine, and 20% methanol). Then, membranes were blocked for 1 h in albumin 5% in TBS + Tween 20. The following primary antibodies were incubated overnight at 4 °C: anti-eNOS (1:500); anti-p-eNOS (1:500); anti-AKT (1:500); anti-p-AKT (1:500); anti-β-actin (1:1000). Membranes were washed three times with wash buffer TBS + Tween-20 and incubated for 1 h at room temperature with the secondary antibody goat anti-mouse IgG secondary antibody, HRP (1:10.000). Membranes were washed three times, and the enhanced chemiluminescence method (Amersham Biosciences, Barcelona, Spain) was used for antibody detection. Autoradiography signals were analyzed with digital image system ImageQuant LAS 4000 (GE Healthcare) (NIH Image, National Institutes of Health, Bethesda, MD, USA).

### 4.9. Drugs

The drugs used in this study are:

Potassium Chloride (KCl, Merck, Darmstadt, Alemania), IN, RN, Acetylcholine, Sodium Nitroprusside, Noradrenaline Hydrochloride, Prazosin, Tetrodotoxin, Guanethidine, Endothelin-1, Indomethacin, L-NAME, S961, BQ123, BQ788 (Sigma-Aldrich, Madrid, Spain). Concentrated drug solutions were made with bi-distilled water, except for indomethacin and prazosin, which were dissolved in ethanol.

### 4.10. Statistical Methods

For organ bath experiments, values are expressed as mean ± standard error of mean, and for Western blot, values are expressed as mean ± SD. Relaxation was expressed as a percentage of inhibition of the agonist-induced response. The contraction was expressed as a percentage of the response to KCl 60 mM. EC50 values (concentration of agonist producing half-maximum effect) were expressed as pD2 (−log EC50) (negative logarithm of the molar concentration at which half-maximum response occurs). The normality of the data was tested with the Shapiro–Wilk test. For intergroup comparisons following a normal distribution, a one-way ANOVA was used. For multiple comparisons (post hoc), the Bonferroni test has been applied. For the concentration-response curves, the comparison has been made on the representative parameters of the curves in the different experimental conditions, pD2 (−log EC50), and the maximum effect (Emax) using a two-way ANOVA. For EFS experiments and Western blot analysis, in which the same aortic rings were used in control and experimental conditions, a paired *t*-test was used. The level of significance used was 5% (*p* < 0.05). Statistical analysis was performed using GraphPad Prism 8.3.0 (GraphPad Software, San Diego, CA, USA).

## 5. Conclusions

RN facilitates the effects of rabbit aortic IN by increasing the sensitivity of vascular wall cells to the hormone. Facilitating the phosphorylation of AKT and eNOS increases the direct effects of IN and those induced by acetylcholine, also increasing the sensitivity of smooth muscle to NO. In addition, RN facilitates the inhibitory effect of IN on vasoconstriction of adrenergic origin, both nervous and due to noradrenaline. The various drugs used in the control of type II diabetes typically focus on increasing the release of IN from the pancreas, which lowers blood glucose and glycosylated hemoglobin. However, there are no drugs for antidiabetic use that have been proven to be effective in reducing vascular resistance to insulin, the most likely origin of type 2 diabetes mellitus. Therefore, RN could be a useful drug in the treatment of this type of diabetes.

## Figures and Tables

**Figure 1 ijms-24-13532-f001:**
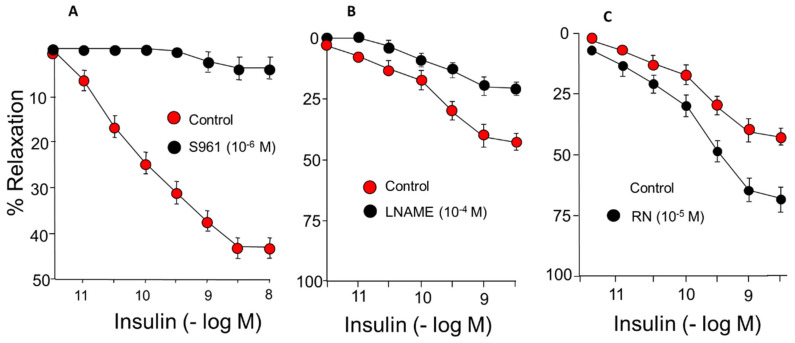
Concentration-response curves to IN (10^−12^–10^−8^ M) in rabbit aorta. (**A**) Effect of S961 (10^−6^ M), IN receptor blocker. (**B**) Effect of L-N^G^-Nitro-L-arginine methyl ester (L-NAME) (10^−4^ M). (**C**) Effect of RN (10^−5^ M).

**Figure 2 ijms-24-13532-f002:**
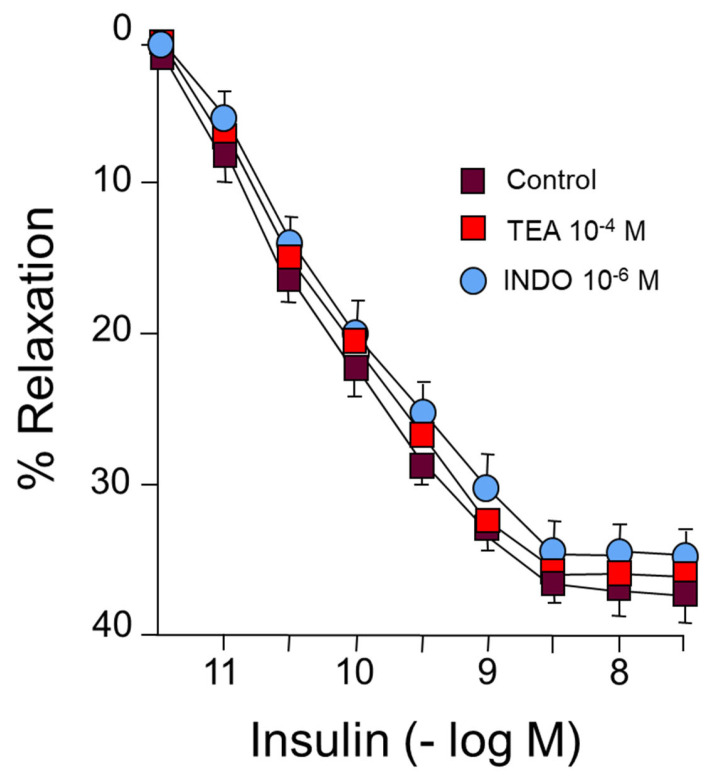
Concentration-response curves to IN (10^−12^–10^−8^ M) in rabbit aorta. A: Effect of tetraethylammonium (TEA) (10^−4^ M) or indomethacin (INDO) (10^−6^ M).

**Figure 3 ijms-24-13532-f003:**
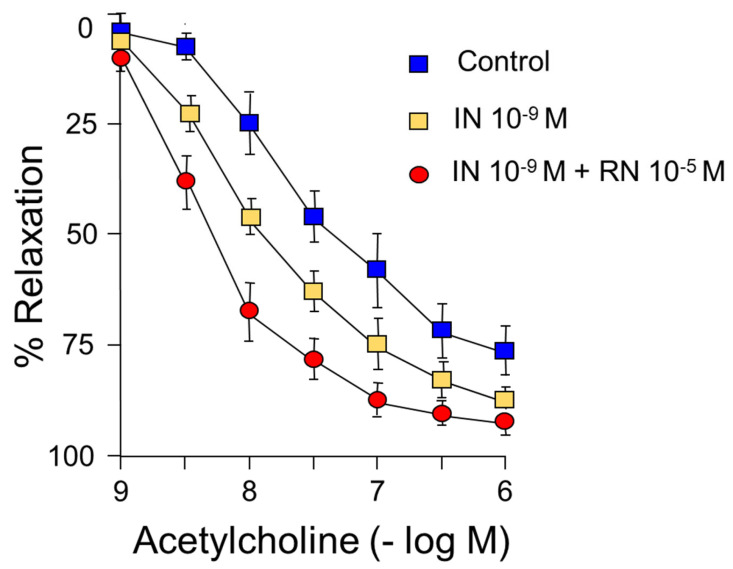
Concentration-response curves to acetylcholine (10^−9^–10^−6^ M). Facilitating effect of IN (10^−9^ M) and IN (10^−9^ M) + RN (10^−5^ M).

**Figure 4 ijms-24-13532-f004:**
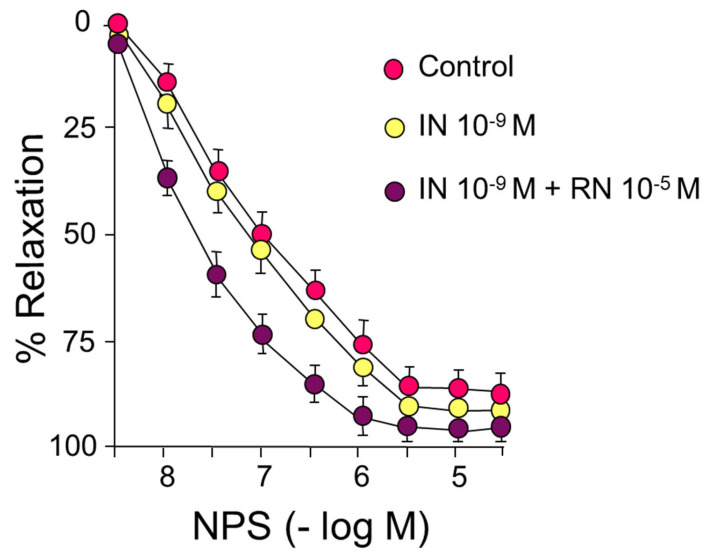
Concentration-response curves to sodium nitroprusside (NPS) (10^−9^–3 × 10^−5^ M). Effects of IN (10^−9^ M) and IN (10^−9^ M) + RN (10^−5^ M).

**Figure 5 ijms-24-13532-f005:**
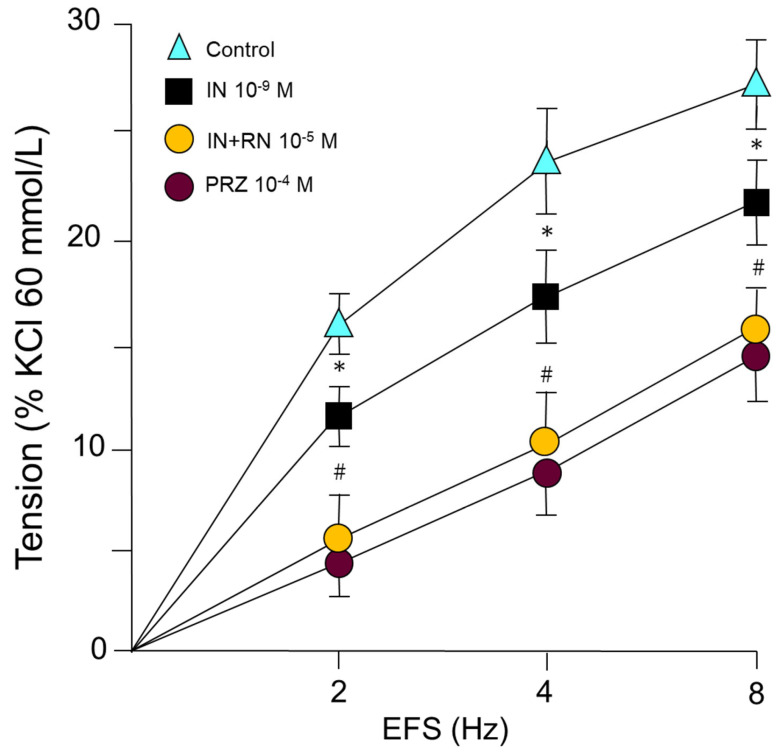
Contractile response induced by electrical field stimulation (EFS) (2, 4 and 8 Hz) in rabbit aorta in the presence of IN (10^−9^ M), IN (10^−9^ M) + RN (10^−5^ M) or PRZ (10^−4^ M). * *p* < 0.05 compared to control. # *p* < 0.05 compared to insulin treated group. *n* = 8 rabbits for each condition.

**Figure 6 ijms-24-13532-f006:**
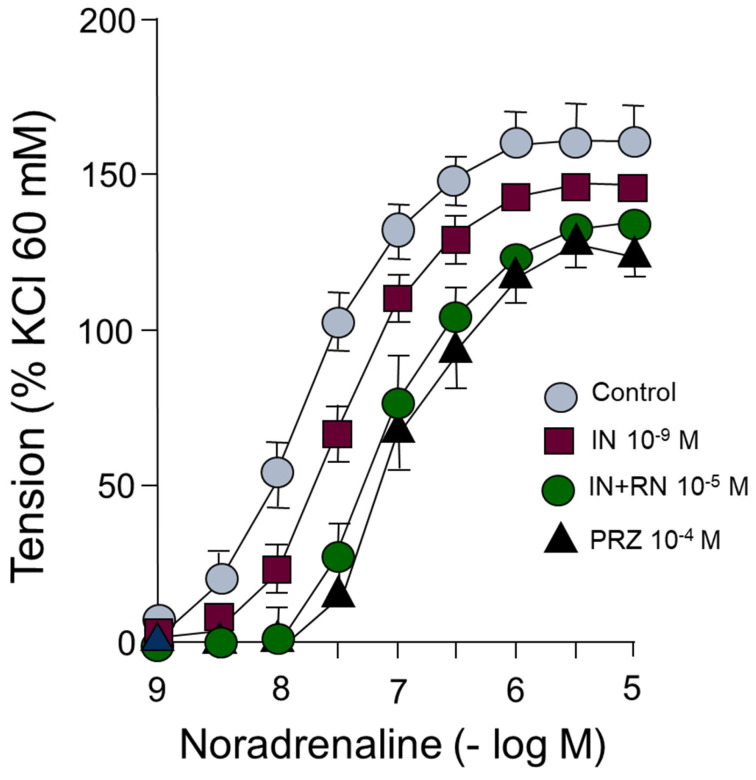
Concentration-response curves to noradrenaline (10^−9^–10^−5^ M) in rabbit aorta in the presence of IN (10^−9^ M), IN (10^−9^ M) + RN (10^−5^ M) or PRZ (10^−4^ M).

**Figure 7 ijms-24-13532-f007:**
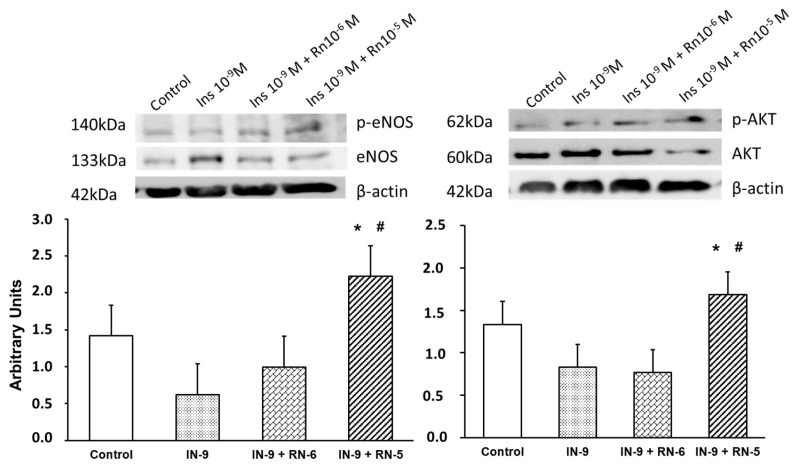
Protein expression levels of eNOS, p-eNOS, AKT and p-AKT in rabbit aorta from control, IN (10^−9^ M), IN + RN (10^−6^), IN + RN (10^−5^) were indicated. The ratio of p-eNOS/eNOS and pAKT/AKT was plotted. Representative western blots for each antibody are added and β-actina was used as internal control. Data are mean ± SD of 5 independent experiments. * *p* < 0.05 vs. control. ^#^
*p* < 0.05 vs. IN group.

**Table 1 ijms-24-13532-t001:** Values of pD_2_ and of the maximum effect (Emax expressed as a percentage of the contraction induced by 60 mM KCl) ± standard error (s.e.), in rabbit aortic segments in the absence (control) and in the presence of the different treatments. * *p* < 0.05 compared to control. *n*: number of rabbits.

IN	*n*	pD_2_ ± e.e	Emax ± e.e
Control	7	9.7 ± 1.0	48 ± 2
S961 10^−6^ M	6	9.6 ± 0.8	5 ± 1 *
LNAME 10^−4^ M	7	8.8 ± 0.3 *	23 ± 2 *
TEA 10^−4^ M	6	9.4 ± 0.4	44 ± 2
BQ123 10^−6^ M	5	9.8 ± 0.3	51 ± 4
BQ788 10^−6^ M	5	9.6 ± 0.2	49 ± 3
INDO 10^−6^ M	6	9.5 ± 0.4	45 ± 3
RN 10^−5^ M	7	10.5 ± 0.2 *	72 ± 3 *

**Table 2 ijms-24-13532-t002:** Values of pD_2_ and of the maximum effect (Emax expressed as a percentage of the contraction induced by 60 mM KCl) ± standard error (s.e.) in rabbit aortic segments in the absence (control) and in the presence of the different treatments. * *p* < 0.05 compared to control. **^#^**
*p* < 0.05 compared to insulin treated group. *n*: number of rabbits.

Acetylcholine	*n*	pD_2_ ± e.e.	Emax ± e.e.
Control	7	7.3 ± 0.2	75 ± 2
IN 10^−9^ M	6	7.9 ± 0.1 *	90 ± 4 *
IN 10^−9^ M + RN 10^−5^ M	6	8.4 ± 0.1 *^#^	91 ± 5 *

**Table 3 ijms-24-13532-t003:** Values of pD2 and of the maximum effect (Emax expressed as a percentage of the contraction induced by 60 mM KCl) ± standard error (s.e.) (NPS), in rabbit aortic segments in the absence (control) and in the presence of the different treatments. * *p* < 0.05 compared to control. ^#^
*p* < 0.05 compared to insulin treated group. *n*: number of rabbits.

NPS	*n*	pD_2_ ± e.e.	Emax ± e.e.
Control	7	7.01 ± 0.24	90 ± 3
IN 10^−9^ M	6	7.03 ± 0.14	92 ± 4
IN 10^−9^ M + RN 10^−5^ M	6	7.61 ± 0.22 *^#^	93 ± 2

**Table 4 ijms-24-13532-t004:** Values of pD_2_ and of the maximum effect (Emax expressed as a percentage of the contraction induced by 60 mM KCl) ± standard error (s.e.), in segments of rabbit aorta in the absence of (control) and in the presence of the different treatments. * *p* < 0.05 compared to control. ^#^
*p* < 0.05 compared to insulin treated group. *n*: number of rabbits.

Noradrenaline	*n*	pD_2_ ± e.e.	Emax ± e.e.
Control	6	7.21 ± 0.23	164.28 ± 5.1
IN 10^−9^ M	7	6.7 ± 0.15 *	155.48 ± 8.7
IN 10^−9^ M + RN 10^−5^ M	6	6.1 ± 0.17 *^#^	152.85 ± 9.2
PRZ 10^−4^ M	5	5.9 ± 0.28 *^#^	148.99 ± 8.3

## Data Availability

Not applicable.

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
