# Peer review of "Improvement of Vascular Insulin Sensitivity by Ranolazine"

_ijms, 2023, doi:10.3390/ijms241713532_

Round 1

Reviewer 1 Report

The study addresses an important issue of cardiovascular consequences of insulin resistance in type 2 diabetes.  It explores the mechanism of the drug  ranozoline, used to treat angina which is also known to improve outcomes for T2D.  The study uses classical pharmacological methods utilising contraction and relaxation of aortic tissue to assess the interaction of insulin and ranolozine in facilitating relaxation, interpreted as a proxy for beneficial, vasodilatory effects. Using combinations of known activators and inhibitors of relevant signalling pathways, the mechanism is explored.

The methodology is robust and the variations between experimental replicates indicate good reproducibility.

Data are presented very clearly, with good use of colour to allow easy identification of different treatments.

The conclusions from the study are described clearly and provide some insight into the mechanism of action of ranolazine.

The description is some parts is unclear, and the following suggestions are intended to facilitate clarification:

Line 14-15 "without changes that induced by sodium nitroprusside" needs re-writing. 

Lines 30-36 describe findings from references 4 and 5, but they need to be interpreted for readers without a strong background in electrophysiology - an extra sentence to link the descriptions of Ca2+ behaviour to vasodilation?

Line 44: the sentence "They would not change in any way" is confusing and can be deleted.

Lines 45-46 - reference 7 is about tea and insulin resistance; needs replacing with something that supports the statement in these lines, especially the claim that the CV system is most sensitive to the effects of insulin.....

Lines 67-69  - ref 28 clearly states an alternative site of action for ranozoline, sodium channels on pancreatic alpha cells, which decreases glucagon production helps to improve post-prandial glucose absorption.  The authors don't mention this anywhere in the paper and suggest that ranozoline's site of action is only vasculature.  Other sites of action should be mentioned.

In the discussion section, when data from the study is described the Figure or Table should be cited.

Lines 186-187 the authors state that the relaxation is mainly due to NO, but data in Figure 1 and Table 1 and line 80 suggest a 45% contribution; needs clarifying.

Line 307-315 - incubation times should be stated

Line 312 needs re-writing - "obtaining" = response recorded?

Line 329 "performing below" awkward phrase and Figure number should be cited instead of "below"  e.g. "shown in Figure X"

Some phrases are awkward or unclear.  Some examples are listed in the comments to the authors. Recommend thorough proof reading of the whole manuscript.

Author Response

Dear reviewer, I am including the answers to your questions.

Thank you very much for your comments

The study addresses an important issue of cardiovascular consequences of insulin resistance in type 2 diabetes.  It explores the mechanism of the drug ranozoline, used to treat angina which is also known to improve outcomes for T2D.  The study uses classical pharmacological methods utilising contraction and relaxation of aortic tissue to assess the interaction of insulin and ranolozine in facilitating relaxation, interpreted as a proxy for beneficial, vasodilatory effects. Using combinations of known activators and inhibitors of relevant signalling pathways, the mechanism is explored.

The methodology is robust and the variations between experimental replicates indicate good reproducibility.

Data are presented very clearly, with good use of colour to allow easy identification of different treatments.

The conclusions from the study are described clearly and provide some insight into the mechanism of action of ranolazine.

The description is some parts is unclear, and the following suggestions are intended to facilitate clarification:

Line 14-15 "without changes that induced by sodium nitroprusside" needs re-writing. 

Changed sentence to: On the other hand, IN facilitated the vasodilator effects of acetylcholine but not the vasodilation induced by sodium nitroprusside.

Lines 30-36 describe findings from references 4 and 5, but they need to be interpreted for readers without a strong background in electrophysiology - an extra sentence to link the descriptions of Ca2+ behaviour to vasodilation?

In both ways, either by stimulating the exit of Ca2+ from the cytosol towards the sarcoplasmic reticulum or by preventing the entry of extracellular Ca2+ into the cytosol, relaxation of the vascular smooth muscle is produced, which leads to vasodilation.

Line 44: the sentence "They would not change in any way" is confusing and can be deleted.

The sentence has been removed

Lines 45-46 - reference 7 is about tea and insulin resistance; needs replacing with something that supports the statement in these lines, especially the claim that the CV system is most sensitive to the effects of insulin.....

The reference 7 has been changed by another one

  • Rensing KL, Reuwer AQ, Arsenault BJ, von der Thuüsen JH, Hoekstra JB, Kastelein JJ, et al. Reducing cardiovascular disease risk in patients with type 2 diabetes and concomitant macrovascular disease: can insulin be too much of a good thing? Diabetes Obes Metab 2011;13:1073-87

Lines 67-69  - ref 28 clearly states an alternative site of action for ranozoline, sodium channels on pancreatic alpha cells, which decreases glucagon production helps to improve post-prandial glucose absorption.  The authors don't mention this anywhere in the paper and suggest that ranozoline's site of action is only vasculature.  Other sites of action should be mentioned.

We added

An alternative site of action for RN, through opening sodium channels, causes a decrease in the release of glucagon by pancreatic alpha cells which improves both postprandial and preprandial glucose levels.

In the discussion section, when data from the study is described the Figure or Table should be cited.

Figure or Table were added in discussion

Lines 186-187 the authors state that the relaxation is mainly due to NO, but data in Figure 1 and Table 1 and line 80 suggest a 45% contribution; needs clarifying.

We added the next sentence

Although in our study NO was responsible for 45% of the relaxation induced by IN. The rest of the vasodilator effect of IN is due to other factors.

Line 307-315 - incubation times should be stated

We add the incubation times corresponding to each experiment

Line 312 needs re-writing - "obtaining" = response recorded?

We re-writing

and then the vasodilator responses to the hormone were recorded

Line 329 "performing below" awkward phrase and Figure number should be cited instead of 

We changed to

 as shown in figure 1, performing the concentration-response curves to IN (Figure 1)

Comments on the Quality of English Language

Some phrases are awkward or unclear.  Some examples are listed in the comments to the authors. Recommend thorough proof reading of the whole manuscript.

The full manuscript has been reviewed for improvement

Reviewer 2 Report

Dear Authors.

Very interesting work and worth publishing. I only have a few comments:

1. not all abbreviations are explained - eg. at the beginning of the Introduction there is no description for NO. In addition, they should be unified - sometimes it says type II diabetic patients and sometimes type 2 diabetes mellitus (as well as in conclusions).

2. in the Introduction (line 40) - type 2 diabetes is the a part of metabolic syndrome so replace and with including 

3. in the Introduction (line 66) - the authors state that different multicenter studies have demonstrated the antidiabetic effects of the drug - it is worth writing whether and how much this drug affects fasting or postprandial glycemia and the percentage of glycated hemoglobin.

4. In the Introduction, it would be good to indicate why you chose the aorta for examination (line 70).

5. in Results - tables - not all abbreviations are explained (e.g. what is pD2?)

6. in conclusions (line 452) - However, there are no drugs for antidiabetic use that are capable of reducing vascular resistance to NI, the origin of type II diabetes - maybe it's better to write that such properties have not been proven.

No

Author Response

Dear reviewer, I am including the answers to your questions.

Thank you very much for your comments.

Dear Authors.

Very interesting work and worth publishing. I only have a few comments:

  1. not all abbreviations are explained - eg. at the beginning of the Introduction there is no description for NO. In addition, they should be unified - sometimes it says type II diabetic patients and sometimes type 2 diabetes mellitus (as well as in conclusions).

Description for NO was added. We unified to type 2 diabetes mellitus

  1. in the Introduction (line 40) - type 2 diabetes is the a part of metabolic syndrome so replace and with including 

We replaced the phrase to Type 2 diabetes mellitus, as part of this syndrome.

  1. in the Introduction (line 66) - the authors state that different multicenter studies have demonstrated the antidiabetic effects of the drug - it is worth writing whether and how much this drug affects fasting or postprandial glycemia and the percentage of glycated hemoglobin.

We added

In type 2 diabetic patients, NR decreased glycosylated hemoglobin (HbA1c) by 7% and reduced the fasting glucose by 25.7 mg/dL.

  1. In the Introduction, it would be good to indicate why you chose the aorta for examination (line 70).

This sentence has been added:

the common arterial vessel that initiates the entire systemic circulation.

  1. in Results - tables - not all abbreviations are explained (e.g. what is pD2?)

pD2: Negative logarithm of the molar concentration at which half-maximum response occurs

  1. in conclusions (line 452) - However, there are no drugs for antidiabetic use that are capable of reducing vascular resistance to NI, the origin of type II diabetes - maybe it's better to write that such properties have not been proven.

The sentence has been changed for the one suggested by the referee.

there are no drugs for antidiabetic use that have been proven to be effective in reducing vascular resistance to insulin, the most likely origin of type 2 diabetes mellitus.

Comments on the Quality of English Language

No
